# Communication-Efficient Distributed Dual Coordinate Ascent

**Martin Jaggi** [*]
ETH Zurich

**Virginia Smith** [*]
UC Berkeley

**Martin Takáč**
Lehigh University

**Jonathan Terhorst**
UC Berkeley

**Sanjay Krishnan**
UC Berkeley

**Thomas Hofmann**
ETH Zurich

**Michael I. Jordan**
UC Berkeley

## Abstract

Communication remains the most significant bottleneck in the performance of distributed optimization algorithms for large-scale machine learning. In this paper, we propose a communication-efficient framework, CoCoA, that uses local computation in a primal-dual setting to dramatically reduce the amount of necessary communication. We provide a strong convergence rate analysis for this class of algorithms, as well as experiments on real-world distributed datasets with implementations in Spark. In our experiments, we find that as compared to state-of-the-art mini-batch versions of SGD and SDCA algorithms, CoCoA converges to the same .001-accurate solution quality on average $25\times$ as quickly.

## 1   Introduction

With the immense growth of available data, developing distributed algorithms for machine learning is increasingly important, and yet remains a challenging topic both theoretically and in practice. On typical real-world systems, communicating data between machines is vastly more expensive than reading data from main memory, e.g. by a factor of several orders of magnitude when leveraging commodity hardware.[1] Yet, despite this reality, most existing distributed optimization methods for machine learning require significant communication between workers, often equalling the amount of local computation (or reading of local data). This includes for example popular mini-batch versions of online methods, such as stochastic subgradient (SGD) and coordinate descent (SDCA).

In this work, we target this bottleneck. We propose a distributed optimization framework that allows one to freely steer the trade-off between *communication* and *local computation*. In doing so, the framework can be easily adapted to the diverse spectrum of available large-scale computing systems, from high-latency commodity clusters to low-latency supercomputers or the multi-core setting.

Our new framework, CoCoA (**Co**mmunication-efficient distributed dual **Co**ordinate **A**scent), supports objectives for linear reguarlized loss minimization, encompassing a broad class of machine learning models. By leveraging the primal-dual structure of these optimization problems, CoCoA effectively combines partial results from local computation while avoiding conflict with updates simultaneously computed on other machines. In each round, CoCoA employs steps of an arbitrary dual optimization method on the local data on each machine, in parallel. A single update vector is then communicated to the master node. For example, when choosing to perform $H$ iterations (usually order of the data size $n$) of an online optimization method locally per round, our scheme saves a factor of $H$ in terms of communication compared to the corresponding naive distributed update

---

[*]Both authors contributed equally.

[1]On typical computers, the latency for accessing data in main memory is in the order of 100 nanoseconds. In contrast, the latency for sending data over a standard network connection is around 250,000 nanoseconds.

scheme (i.e., updating a single point before communication). When processing the same number of datapoints, this is clearly a dramatic savings.

Our theoretical analysis (Section 4) shows that this significant reduction in communication cost comes with only a very moderate increase in the amount of total computation, in order to reach the same optimization accuracy. We show that, in general, the distributed CoCoA framework will inherit the convergence rate of the internally-used local optimization method. When using SDCA (randomized dual coordinate ascent) as the local optimizer and assuming smooth losses, this convergence rate is geometric.

In practice, our experiments with the method implemented on the fault-tolerant Spark platform [1] confirm both the clock time performance and huge communication savings of the proposed method on a variety distributed datasets. Our experiments consistently show order of magnitude gains over traditional mini-batch methods of both SGD and SDCA, and significant gains over the faster but theoretically less justified local SGD methods.

**Related Work.** As we discuss below (Section 5), our approach is distinguished from recent work on parallel and distributed optimization [2, 3, 4, 5, 6, 7, 8, 9] in that we provide a general framework for improving the communication efficiency of *any* dual optimization method. To the best of our knowledge, our work is the first to analyze the convergence rate for an algorithm with this level of communication efficiency, without making data-dependent assumptions. The presented analysis covers the case of smooth losses, but should also be extendable to the non-smooth case. Existing methods using mini-batches [4, 2, 10] are closely related, though our algorithm makes significant improvements by immediately applying all updates locally while they are processed, a scheme that is not considered in the classic mini-batch setting. This intuitive modification results in dramatically improved empirical results and also strengthens our theoretical convergence rate. More precisely, the convergence rate shown here only degrades with the number of workers $K$, instead of with the significantly larger mini-batch-size (typically order $n$) in the case of mini-batch methods.

Our method builds on a closely related recent line of work of [2, 3, 11, 12]. We generalize the algorithm of [2, 3] by allowing the use of arbitrary (dual) optimization methods as the local subroutine within our framework. In the special case of using coordinate ascent as the local optimizer, the resulting algorithm is very similar, though with a different computation of the coordinate updates. Moreover, we provide the first theoretical convergence rate analysis for such methods, without making strong assumptions on the data.

The proposed CoCoA framework in its basic variant is entirely free of tuning parameters or learning rates, in contrast to SGD-based methods. The only choice to make is the selection of the internal local optimization procedure, steering the desired trade-off between communication and computation. When choosing a primal-dual optimizer as the internal procedure, the duality gap readily provides a fair stopping criterion and efficient accuracy certificates during optimization.

**Paper Outline.** The rest of the paper is organized as follows. In Section 2 we describe the problem setting of interest. Section 3 outlines the proposed framework, CoCoA, and the convergence analysis of this method is presented in Section 4. We discuss related work in Section 5, and compare against several other state-of-the-art methods empirically in Section 6.

## 2 Setup

A large class of methods in machine learning and signal processing can be posed as the minimization of a convex loss function of linear predictors with a convex regularization term:

$$\min_{w \in \mathbb{R}^d} \quad \left[ P(\boldsymbol{w}) := \frac{\lambda}{2} \|\boldsymbol{w}\|^2 + \frac{1}{n} \sum_{i=1}^{n} \ell_i(\boldsymbol{w}^T \boldsymbol{x}_i) \right], \tag{1}$$

Here the data training examples are real-valued vectors $\boldsymbol{x}_i \in \mathbb{R}^d$; the loss functions $\ell_i, i = 1, \ldots, n$ are convex and depend possibly on labels $y_i \in \mathbb{R}$; and $\lambda > 0$ is the regularization parameter. Using the setup of [13], we assume the regularizer is the $\ell_2$-norm for convenience. Examples of this class of problems include support vector machines, as well as regularized linear and logistic regression, ordinal regression, and others.

The most popular method to solve problems of the form (1) is the *stochastic subgradient method* (SGD) [14, 15, 16]. In this setting, SGD becomes an online method where every iteration only requires access to a single data example $(\boldsymbol{x}_i, y_i)$, and the convergence rate is well-understood.

The associated conjugate *dual* problem of (1) takes the following form, and is defined over one dual variable per each example in the training set.

$$\max_{\boldsymbol{\alpha} \in \mathbb{R}^n} \left[ D(\boldsymbol{\alpha}) := -\frac{\lambda}{2} \|A\boldsymbol{\alpha}\|^2 - \frac{1}{n} \sum_{i=1}^{n} \ell_i^*(-\alpha_i) \right], \tag{2}$$

where $\ell_i^*$ is the conjugate (Fenchel dual) of the loss function $\ell_i$, and the data matrix $A \in \mathbb{R}^{d \times n}$ collects the (normalized) data examples $A_i := \frac{1}{\lambda n} \boldsymbol{x}_i$ in its columns. The duality comes with the convenient mapping from dual to primal variables $\boldsymbol{w}(\boldsymbol{\alpha}) := A\boldsymbol{\alpha}$ as given by the optimality conditions [13]. For any configuration of the dual variables $\boldsymbol{\alpha}$, we have the duality gap defined as $P(\boldsymbol{w}(\boldsymbol{\alpha})) - D(\boldsymbol{\alpha})$. This gap is a computable certificate of the approximation quality to the unknown true optimum $P(\boldsymbol{w}^*) = D(\boldsymbol{\alpha}^*)$, and therefore serves as a useful stopping criteria for algorithms.

For problems of the form (2), *coordinate descent* methods have proven to be very efficient, and come with several benefits over primal methods. In randomized *dual coordinate ascent* (SDCA), updates are made to the dual objective (2) by solving for one coordinate completely while keeping all others fixed. This algorithm has been implemented in a number of software packages (e.g. LibLinear [17]), and has proven very suitable for use in large-scale problems, while giving stronger convergence results than the primal-only methods (such as SGD), at the same iteration cost [13]. In addition to superior performance, this method also benefits from requiring no stepsize, and having a well-defined stopping criterion given by the duality gap.

## 3  Method Description

The COCOA framework, as presented in Algorithm 1, assumes that the data $\{(\boldsymbol{x}_i, y_i)\}_{i=1}^n$ for a regularized loss minimization problem of the form (1) is distributed over $K$ worker machines. We associate with the datapoints their corresponding dual variables $\{\alpha_i\}_{i=1}^n$, being partitioned between the workers in the same way. The core idea is to use the dual variables to efficiently merge the parallel updates from the different workers without much conflict, by exploiting the fact that they all work on disjoint sets of dual variables.

---

**Algorithm 1:** COCOA: Communication-Efficient Distributed Dual Coordinate Ascent

**Input**: $T \geq 1$, scaling parameter $1 \leq \beta_K \leq K$ (default: $\beta_K := 1$).
**Data**: $\{(\boldsymbol{x}_i, y_i)\}_{i=1}^n$ distributed over $K$ machines
**Initialize**: $\boldsymbol{\alpha}_{[k]}^{(0)} \leftarrow \mathbf{0}$ for all machines $k$, and $\boldsymbol{w}^{(0)} \leftarrow \mathbf{0}$
**for** $t = 1, 2, \ldots, T$
    **for** *all machines* $k = 1, 2, \ldots, K$ *in parallel*
        $(\Delta\boldsymbol{\alpha}_{[k]}, \Delta\boldsymbol{w}_k) \leftarrow$ LOCALDUALMETHOD$(\boldsymbol{\alpha}_{[k]}^{(t-1)}, \boldsymbol{w}^{(t-1)})$
        $\boldsymbol{\alpha}_{[k]}^{(t)} \leftarrow \boldsymbol{\alpha}_{[k]}^{(t-1)} + \frac{\beta_K}{K}\Delta\boldsymbol{\alpha}_{[k]}$
    **end**
    *reduce* $\boldsymbol{w}^{(t)} \leftarrow \boldsymbol{w}^{(t-1)} + \frac{\beta_K}{K} \sum_{k=1}^{K} \Delta\boldsymbol{w}_k$
**end**

---

In each round, the $K$ workers in parallel perform some steps of an arbitrary optimization method, applied to their local data. This internal procedure tries to maximize the dual formulation (2), only with respect to their own local dual variables. We call this local procedure LOCALDUALMETHOD, as specified in the template Procedure A. Our core observation is that the necessary information each worker requires about the state of the other dual variables can be very compactly represented by a single primal vector $\boldsymbol{w} \in \mathbb{R}^d$, without ever sending around data or dual variables between the machines.

Allowing the subroutine to process more than one local data example per round dramatically reduces the amount of communication between the workers. By definition, COCOA in each outer iteration

**Procedure A:** LOCALDUALMETHOD: Dual algorithm for prob. (2) on a single coordinate block $k$

---

**Input**: Local $\boldsymbol{\alpha}_{[k]} \in \mathbb{R}^{n_k}$, and $\boldsymbol{w} \in \mathbb{R}^d$ consistent with other coordinate blocks of $\boldsymbol{\alpha}$ s.t. $\boldsymbol{w} = A\boldsymbol{\alpha}$
**Data**: Local $\{(\boldsymbol{x}_i, y_i)\}_{i=1}^{n_k}$
**Output**: $\Delta\boldsymbol{\alpha}_{[k]}$ and $\Delta\boldsymbol{w} := A_{[k]}\Delta\boldsymbol{\alpha}_{[k]}$

---

**Procedure B:** LOCALSDCA: SDCA iterations for problem (2) on a single coordinate block $k$

---

**Input**: $H \geq 1$, $\boldsymbol{\alpha}_{[k]} \in \mathbb{R}^{n_k}$, and $\boldsymbol{w} \in \mathbb{R}^d$ consistent with other coordinate blocks of $\boldsymbol{\alpha}$ s.t. $\boldsymbol{w} = A\boldsymbol{\alpha}$
**Data**: Local $\{(\boldsymbol{x}_i, y_i)\}_{i=1}^{n_k}$
**Initialize**: $\boldsymbol{w}^{(0)} \leftarrow \boldsymbol{w}$, $\Delta\boldsymbol{\alpha}_{[k]} \leftarrow \boldsymbol{0} \in \mathbb{R}^{n_k}$
**for** $h = 1, 2, \ldots, H$
  *choose $i \in \{1, 2, \ldots, n_k\}$ uniformly at random*
  *find $\Delta\alpha$ maximizing* $-\frac{\lambda n}{2}\|\boldsymbol{w}^{(h-1)} + \frac{1}{\lambda n}\Delta\alpha\,\boldsymbol{x}_i\|^2 - \ell_i^*\big(-(\alpha_i^{(h-1)} + \Delta\alpha)\big)$
  $\alpha_i^{(h)} \leftarrow \alpha_i^{(h-1)} + \Delta\alpha$
  $(\Delta\boldsymbol{\alpha}_{[k]})_i \leftarrow (\Delta\boldsymbol{\alpha}_{[k]})_i + \Delta\alpha$
  $\boldsymbol{w}^{(h)} \leftarrow \boldsymbol{w}^{(h-1)} + \frac{1}{\lambda n}\Delta\alpha\,\boldsymbol{x}_i$
**end**
**Output**: $\Delta\boldsymbol{\alpha}_{[k]}$ and $\Delta\boldsymbol{w} := A_{[k]}\Delta\boldsymbol{\alpha}_{[k]}$

---

only requires communication of a single vector for each worker, that is $\Delta\boldsymbol{w}_k \in \mathbb{R}^d$. Further, as we will show in Section 4, COCOA inherits the convergence guarantee of any algorithm run locally on each node in the inner loop of Algorithm 1. We suggest to use randomized dual coordinate ascent (SDCA) [13] as the internal optimizer in practice, as implemented in Procedure B, and also used in our experiments.

**Notation.** In the same way the data is partitioned across the $K$ worker machines, we write the dual variable vector as $\boldsymbol{\alpha} = (\boldsymbol{\alpha}_{[1]}, \ldots, \boldsymbol{\alpha}_{[K]}) \in \mathbb{R}^n$ with the corresponding coordinate blocks $\boldsymbol{\alpha}_{[k]} \in \mathbb{R}^{n_k}$ such that $\sum_k n_k = n$. The submatrix $A_{[k]}$ collects the columns of $A$ (i.e. rescaled data examples) which are available locally on the $k$-th worker. The parameter $T$ determines the number of outer iterations of the algorithm, while when using an online internal method such as LOCALSDCA, then the number of inner iterations $H$ determines the computation-communication trade-off factor.

## 4 Convergence Analysis

Considering the dual problem (2), we define the local suboptimality on each coordinate block as:

$$\varepsilon_{D,k}(\boldsymbol{\alpha}) := \max_{\hat{\boldsymbol{\alpha}}_{[k]} \in \mathbb{R}^{n_k}} D((\boldsymbol{\alpha}_{[1]}, \ldots, \hat{\boldsymbol{\alpha}}_{[k]}, \ldots, \boldsymbol{\alpha}_{[K]})) - D((\boldsymbol{\alpha}_{[1]}, \ldots, \boldsymbol{\alpha}_{[k]}, \ldots, \boldsymbol{\alpha}_{[K]})), \quad (3)$$

that is how far we are from the optimum on block $k$ with all other blocks fixed. Note that this differs from the global suboptimality $\max_{\hat{\boldsymbol{\alpha}}} D(\hat{\boldsymbol{\alpha}}) - D((\boldsymbol{\alpha}_{[1]}, \ldots, \boldsymbol{\alpha}_{[K]}))$.

**Assumption 1** (Local Geometric Improvement of LOCALDUALMETHOD). *We assume that there exists $\Theta \in [0, 1)$ such that for any given $\boldsymbol{\alpha}$, LOCALDUALMETHOD when run on block $k$ alone returns a (possibly random) update $\Delta\boldsymbol{\alpha}_{[k]}$ such that*

$$\mathbf{E}[\epsilon_{D,k}((\boldsymbol{\alpha}_{[1]}, \ldots, \boldsymbol{\alpha}_{[k-1]}, \boldsymbol{\alpha}_{[k]} + \Delta\boldsymbol{\alpha}_{[k]}, \boldsymbol{\alpha}_{[k+1]}, \ldots, \boldsymbol{\alpha}_{[K]}))] \leq \Theta \cdot \epsilon_{D,k}(\boldsymbol{\alpha}). \quad (4)$$

Note that this assumption is satisfied for several available implementations of the inner procedure LOCALDUALMETHOD, in particular for LOCALSDCA, as shown in the following Proposition.

From here on, we assume that the input data is scaled such that $\|\boldsymbol{x}_i\| \leq 1$ for all datapoints. Proofs of all statements are provided in the supplementary material.

**Proposition 1.** *Assume the loss functions $\ell_i$ are $(1/\gamma)$-smooth. Then for using LOCALSDCA, Assumption 1 holds with*

$$\Theta = \left(1 - \frac{\lambda n\gamma}{1 + \lambda n\gamma}\frac{1}{\tilde{n}}\right)^H. \quad (5)$$

*where $\tilde{n} := \max_k n_k$ is the size of the largest block of coordinates.*

**Theorem 2.** *Assume that Algorithm 1 is run for $T$ outer iterations on $K$ worker machines, with the procedure* LOCALDUALMETHOD *having local geometric improvement $\Theta$, and let $\beta_K := 1$. Further, assume the loss functions $\ell_i$ are $(1/\gamma)$-smooth. Then the following geometric convergence rate holds for the global (dual) objective:*

$$\mathbf{E}[D(\boldsymbol{\alpha}^*) - D(\boldsymbol{\alpha}^{(T)})] \leq \left(1 - (1 - \Theta)\frac{1}{K}\frac{\lambda n \gamma}{\sigma + \lambda n \gamma}\right)^T \left(D(\boldsymbol{\alpha}^*) - D(\boldsymbol{\alpha}^{(0)})\right). \qquad (6)$$

*Here $\sigma$ is any real number satisfying*

$$\sigma \geq \sigma_{\min} := \max_{\boldsymbol{\alpha} \in \mathbb{R}^n} \lambda^2 n^2 \frac{\sum_{k=1}^{K}\|A_{[k]}\boldsymbol{\alpha}_{[k]}\|^2 - \|A\boldsymbol{\alpha}\|^2}{\|\boldsymbol{\alpha}\|^2} \geq 0. \qquad (7)$$

**Lemma 3.** *If $K = 1$ then $\sigma_{\min} = 0$. For any $K \geq 1$, when assuming $\|\boldsymbol{x}_i\| \leq 1 \ \forall i$, we have*

$$0 \leq \sigma_{\min} \leq \tilde{n}.$$

*Moreover, if datapoints between different workers are orthogonal, i.e. $(A^T A)_{i,j} = 0 \ \forall i, j$ such that $i$ and $j$ do not belong to the same part, then $\sigma_{\min} = 0$.*

If we choose $K = 1$ then, Theorem 2 together with Lemma 3 implies that

$$\mathbf{E}[D(\boldsymbol{\alpha}^*) - D(\boldsymbol{\alpha}^{(T)})] \leq \Theta^T \left(D(\boldsymbol{\alpha}^*) - D(\boldsymbol{\alpha}^{(0)})\right),$$

as expected, showing that the analysis is tight in the special case $K = 1$. More interestingly, we observe that for any $K$, in the extreme case when the subproblems are solved to optimality (i.e. letting $H \to \infty$ in LOCALSDCA), then the algorithm as well as the convergence rate match that of serial/parallel *block-coordinate descent* [18, 19].

**Note:** If choosing the starting point as $\boldsymbol{\alpha}^{(0)} := \mathbf{0}$ as in the main algorithm, then it is known that $D(\boldsymbol{\alpha}^*) - D(\boldsymbol{\alpha}^{(0)}) \leq 1$ (see e.g. Lemma 20 in [13]).

## 5 Related Work

**Distributed Primal-Dual Methods.** Our approach is most closely related to recent work by [2, 3], which generalizes the distributed optimization method for linear SVMs as in [11] to the primal-dual setting considered here (which was introduced by [13]). The difference between our approach and the 'practical' method of [2] is that our internal steps directly correspond to coordinate descent iterations on the global dual objective (2), for coordinates in the current block, while in [3, Equation 8] and [2], the inner iterations apply to a slightly different notion of the sub-dual problem defined on the local data. In terms of convergence results, the analysis of [2] only addresses the mini-batch case without local updates, while the more recent paper [3] shows a convergence rate for a variant of COCOA with inner coordinate steps, but under the unrealistic assumption that the data is orthogonal between the different workers. In this case, the optimization problems become independent, so that an even simpler single-round communication scheme summing the individual resulting models $\boldsymbol{w}$ would give an exact solution. Instead, we show a linear convergence rate for the full problem class of smooth losses, without any assumptions on the data, in the same generality as the non-distributed setting of [13].

While the experimental results in all papers [11, 2, 3] are encouraging for this type of method, they do not yet provide a quantitative comparison of the gains in communication efficiency, or compare to the analogous SGD schemes that use the same distribution and communication patterns, which is the main goal or our experiments in Section 6. For the special case of linear SVMs, the first paper to propose the same algorithmic idea was [11], which used LibLinear in the inner iterations. However, the proposed algorithm [11] processes the blocks sequentially (not in the parallel or distributed setting). Also, it is assumed that the subproblems are solved to near optimality on each block before selecting the next, making the method essentially standard block-coordinate descent. While no convergence rate was given, the empirical results in the journal paper [12] suggest that running LibLinear for just one pass through the local data performs well in practice. Here, we prove this, quantify the communication efficiency, and show that fewer local steps can improve the overall performance. For the LASSO case, [7] has proposed a parallel coordinate descent method converging to the true optimum, which could potentially also be interpreted in our framework here.

**Mini-Batches.** Another closely related avenue of research includes methods that use *mini-batches* to distribute updates. In these methods, a mini-batch, or sample, of the data examples is selected for processing at each iteration. All updates within the mini-batch are computed based on the same fixed parameter vector $w$, and then these updates are either added or averaged in a reduce step and communicated back to the worker machines. This concept has been studied for both SGD and SDCA, see e.g. [4, 10] for the SVM case. The so-called naive variant of [2] is essentially identical to mini-batch dual coordinate descent, with a slight difference in defining the sub-problems.

As is shown in [2] and below in Section 6, the performance of these algorithms suffers when processing large batch sizes, as they do not take local updates immediately into account. Furthermore, they are very sensitive to the choice of the parameter $\beta_b$, which controls the magnitude of combining all updates between $\beta_b := 1$ for (conservatively) averaging, and $\beta_b := b$ for (aggressively) adding the updates (here we denote $b$ as the size of the selected mini-batch, which can be of size up to $n$). This instability is illustrated by the fact that even the change of $\beta_b := 2$ instead of $\beta_b := 1$ can lead to divergence of coordinate descent (SDCA) in the simple case of just two coordinates [4] . In practice it can be very difficult to choose the correct data-dependent parameter $\beta_b$ especially for large mini-batch sizes $b \approx n$, as the parameter range spans many orders of magnitude, and directly controls the step size of the resulting algorithm, and therefore the convergence rate [20, 21]. For sparse data, the work of [20, 21] gives some data dependent choices of $\beta_b$ which are safe.

Known convergence rates for the mini-batch methods degrade linearly with the growing batch size $b \approx \Theta(n)$. More precisely, the improvement in objective function per example processed degrades with a factor of $\beta_b$ in [4, 20, 21]. In contrast, our convergence rate as shown in Theorem 2 only degrades with the much smaller number of worker machines $K$, which in practical applications is often several orders of magnitudes smaller than the mini-batch size $b$.

**Single Round of Communication.** One extreme is to consider methods with only a single round of communication (e.g. one map-reduce operation), as in [22, 6, 23]. The output of these methods is the average of $K$ individual models, trained only on the local data on each machine. In [22], the authors give conditions on the data and computing environment under which these one-communication algorithms may be sufficient. In general, however, the true optimum of the original problem (1) is *not* the average of these $K$ models, no matter how accurately the subproblems are solved [24].

**Naive Distributed Online Methods, Delayed Gradients, and Multi-Core.** On the other extreme, a natural way to distribute updates is to let every machine send updates to the master node (sometimes called the "parameter server") as soon as they are performed. This is what we call the *naive distributed SGD / CD* in our experiments. The amount of communication for such naive distributed online methods is the same as the number of data examples processed. In contrast to this, the number of communicated vectors in our method is divided by $H$, that is the number of inner local steps performed per outer iteration, which can be $\Theta(n)$.

The early work of [25] introduced the nice framework of gradient updates where the gradients come with some delays, i.e. are based on outdated iterates, and shows some robust convergence rates. In the machine learning setting, [26] and the later work of [27] have provided additional insights into these types of methods. However, these papers study the case of smooth objective functions of a sum structure, and so do not directly apply to general case we consider here. In the same spirit, [5] implements SGD with communication-intense updates after each example processed, allowing asynchronous updates again with some delay. For coordinate descent, the analogous approach was studied in [28]. Both methods [5, 28] are $H$ times less efficient in terms of communication when compared to CoCoA, and are designed for *multi-core* shared memory machines (where communication is as fast as memory access). They require the same amount of communication as *naive distributed SGD / CD*, which we include in our experiments in Section 6, and a slightly larger number of iterations due to the asynchronicity. The $1/t$ convergence rate shown in [5] only holds under strong sparsity assumptions on the data. A more recent paper [29] deepens the understanding of such methods, but still only applies to very sparse data. For general data, [30] theoretically shows that $1/\varepsilon^2$ communications rounds of single vectors are enough to obtain $\varepsilon$-quality for linear classifiers, with the rate growing with $K^2$ in the number of workers. Our new analysis here makes the dependence on $1/\varepsilon$ logarithmic.

# 6 Experiments

In this section, we compare CoCoA to traditional mini-batch versions of stochastic dual coordinate ascent and stochastic gradient descent, as well as the locally-updating version of stochastic gradient descent. We implement mini-batch SDCA (denoted mini-batch-CD) as described in [4, 2]. The SGD-based methods are mini-batch and locally-updating versions of Pegasos [16], differing only in whether the primal vector is updated locally on each inner iteration or not, and whether the resulting combination/communication of the updates is by an average over the total size $KH$ of the mini-batch (mini-batch-SGD) or just over the number of machines $K$ (local-SGD). For each algorithm, we additionally study the effect of scaling the average by a parameter $\beta_K$, as first described in [4], while noting that it is a benefit to avoid having to tune this data-dependent parameter.

We apply these algorithms to standard hinge loss $\ell_2$-regularized support vector machines, using implementations written in Spark on m1.large Amazon EC2 instances [1]. Though this non-smooth case is not yet covered in our theoretical analysis, we still see remarkable empirical performance. Our results indicate that CoCoA is able to converge to .001-accurate solutions nearly $25\times$ as fast compared the other algorithms, when all use $\beta_K = 1$. The datasets used in these analyses are summarized in Table 1, and were distributed among $K = 4, 8,$ and 32 nodes, respectively. We use the same regularization parameters as specified in [16, 17].

Table 1: Datasets for Empirical Study

| Dataset | Training $(n)$ | Features $(d)$ | Sparsity | $\lambda$ | Workers $(K)$ |
|---------|---------------|----------------|----------|-----------|----------------|
| cov | 522,911 | 54 | 22.22% | 1e-6 | 4 |
| rcv1 | 677,399 | 47,236 | 0.16% | 1e-6 | 8 |
| imagenet | 32,751 | 160,000 | 100% | 1e-5 | 32 |

In comparing each algorithm and dataset, we analyze progress in primal objective value as a function of both time (Figure 1) and communication (Figure 2). For all competing methods, we present the result for the batch size $(H)$ that yields the best performance in terms of reduction in objective value over time. For the locally-updating methods (CoCoA and local-SGD), these tend to be larger batch sizes corresponding to processing almost all of the local data at each outer step. For the non-locally updating mini-batch methods, (mini-batch SDCA [4] and mini-batch SGD [16]), these typically correspond to smaller values of $H$, as averaging the solutions to guarantee safe convergence becomes less of an impediment for smaller batch sizes.

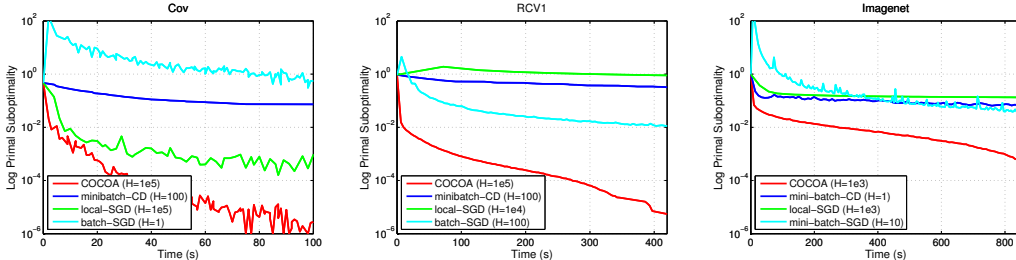

Figure 1: Primal Suboptimality vs. Time for Best Mini-Batch Sizes (H): For $\beta_K = 1$, CoCoA converges more quickly than all other algorithms, even when accounting for different batch sizes.

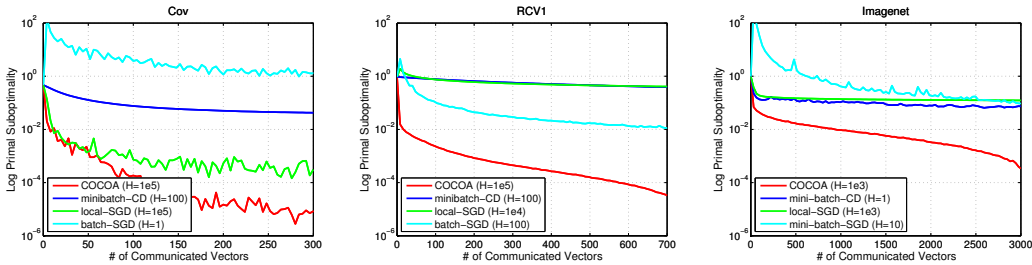

Figure 2: Primal Suboptimality vs. # of Communicated Vectors for Best Mini-Batch Sizes (H): A clear correlation is evident between the number of communicated vectors and wall-time to convergence (Figure 1).

First, we note that there is a clear correlation between the wall-time spent processing each dataset and the number of vectors communicated, indicating that communication has a significant effect on convergence speed. We see clearly that CoCoA is able to converge to a more accurate solution in all datasets much faster than the other methods. On average, CoCoA reaches a .001-accurate solution for these datasets 25x faster than the best competitor. This is a testament to the algorithm's ability to avoid communication while still making significant global progress by efficiently combining the local updates of each iteration. The improvements are robust for both regimes $n \gg d$ and $n \ll d$.

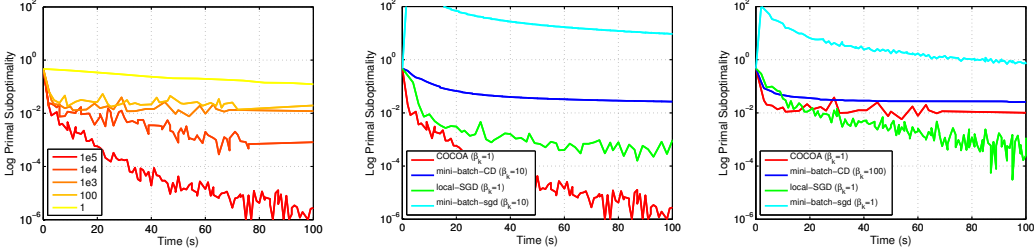

Figure 3: Effect of $H$ on CoCoA.  Figure 4: Best $\beta_K$ Scaling Values for $H = 1e5$ and $H = 100$.

In Figure 3 we explore the effect of $H$, the computation-communication trade-off factor, on the convergence of CoCoA for the Cov dataset on a cluster of 4 nodes. As described above, increasing $H$ decreases communication but also affects the convergence properties of the algorithm. In Figure 4, we attempt to scale the averaging step of each algorithm by using various $\beta_K$ values, for two different batch sizes on the Cov dataset ($H = 1e5$ and $H = 100$). We see that though $\beta_K$ has a larger impact on the smaller batch size, it is still not enough to improve the mini-batch algorithms beyond what is achieved by CoCoA and local-SGD.

## 7 Conclusion

We have presented a communication-efficient framework for distributed dual coordinate ascent algorithms that can be used to solve large-scale regularized loss minimization problems. This is crucial in settings where datasets must be distributed across multiple machines, and where communication amongst nodes is costly. We have shown that the proposed algorithm performs competitively on real-world, large-scale distributed datasets, and have presented the first theoretical analysis of this algorithm that achieves competitive convergence rates without making additional assumptions on the data itself.

It remains open to obtain improved convergence rates for more aggressive updates corresponding to $\beta_K > 1$, which might be suitable for using the 'safe' updates techniques of [4] and the related expected separable over-approximations of [18, 19], here applied to $K$ instead of $n$ blocks. Furthermore, it remains open to show convergence rates for local SGD in the same communication efficient setting as described here.

**Acknowledgments.** We thank Shivaram Venkataraman, Ameet Talwalkar, and Peter Richtárik for fruitful discussions. MJ acknowledges support by the Simons Institute for the Theory of Computing.

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
