[Supplementary Material]

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

# A  Proof of Theorem 2 – Main Convergence Theorem

In the following, for a given vector $\boldsymbol{\alpha}_{[k]} \in \mathbb{R}^{n_k}$, we write $\boldsymbol{\alpha}_{\langle[k]\rangle} \in \mathbb{R}^n$ for its zero-padded version which coincides with $\boldsymbol{\alpha}_{[k]}$ on the $k$-th coordinate block, and is zero everywhere else.

**Theorem' 2.** *Assume that Algorithm 1 is run for $T$ outer iterations on $K$ worker machines, with the procedure* LOCALDUALMETHOD *having local geometric improvement $\Theta$, and let $\beta_K := 1$. Further, assume the loss functions $\ell_i$ are $(1/\gamma)$-smooth. Then the following geometric convergence rate holds for the global (dual) objective:*

$$\mathbf{E}[D(\boldsymbol{\alpha}^*) - D(\boldsymbol{\alpha}^{(T)})] \leq \left(1 - (1 - \Theta)\frac{1}{K}\frac{\lambda n \gamma}{\sigma + \lambda n \gamma}\right)^T \left(D(\boldsymbol{\alpha}^*) - D(\boldsymbol{\alpha}^{(0)})\right).$$

*Here $\sigma$ is any real number satisfying*

$$\sigma \geq \sigma_{\min} := \max_{\boldsymbol{\alpha} \in \mathbb{R}^n} \lambda^2 n^2 \frac{\sum_{k=1}^K \|A_{[k]}\boldsymbol{\alpha}_{[k]}\|^2 - \|A\boldsymbol{\alpha}\|^2}{\|\boldsymbol{\alpha}\|^2} \geq 0.$$

*Proof.* From the definition of the update performed by Algorithm 1 (for the setting of $\beta_K := 1$), we have $\boldsymbol{\alpha}^{(t+1)} = \boldsymbol{\alpha}^{(t)} + \frac{1}{K}\sum_{k=1}^K \Delta\boldsymbol{\alpha}_{\langle[k]\rangle}$. Let us estimate the change of objective function after one outer iteration. Then using concavity of $D$ we have

$$D(\boldsymbol{\alpha}^{(t+1)}) = D\left(\boldsymbol{\alpha}^{(t)} + \frac{1}{K}\sum_{k=1}^K \Delta\boldsymbol{\alpha}_{\langle[k]\rangle}\right) = D\left(\frac{1}{K}\sum_{k=1}^K(\boldsymbol{\alpha}^{(t)} + \Delta\boldsymbol{\alpha}_{\langle[k]\rangle})\right)$$
$$\geq \frac{1}{K}\sum_{k=1}^K D(\boldsymbol{\alpha}^{(t)} + \Delta\boldsymbol{\alpha}_{\langle[k]\rangle}).$$

Subtracting $D(\boldsymbol{\alpha}^{(t)})$ from both sides and denoting by $\hat{\alpha}_{[k]}^*$ the local maximizer as in (3) we obtain

$$D(\boldsymbol{\alpha}^{(t+1)}) - D(\boldsymbol{\alpha}^{(t)}) \geq \frac{1}{K}\sum_{k=1}^K \left[D(\boldsymbol{\alpha}^{(t)} + \Delta\boldsymbol{\alpha}_{\langle[k]\rangle}) - D(\boldsymbol{\alpha}^{(t)})\right]$$
$$= \frac{1}{K}\sum_{k=1}^K \left[D(\boldsymbol{\alpha}^{(t)} + \Delta\boldsymbol{\alpha}_{\langle[k]\rangle}) - D((\boldsymbol{\alpha}_{[t]}^{(1)}, \ldots, \hat{\boldsymbol{\alpha}}_{[k]}^*, \ldots, \boldsymbol{\alpha}_{[t]}^{(K)}))\right.$$
$$\left. + D((\boldsymbol{\alpha}_{[t]}^{(1)}, \ldots, \hat{\boldsymbol{\alpha}}_{[k]}^*, \ldots, \boldsymbol{\alpha}_{[t]}^{(K)})) - D(\boldsymbol{\alpha}^{(t)})\right]$$
$$\overset{(3)}{=} \frac{1}{K}\sum_{k=1}^K \left[\varepsilon_{D,k}(\boldsymbol{\alpha}^{(t)}) - \varepsilon_{D,k}(\boldsymbol{\alpha}^{(t)} + \Delta\boldsymbol{\alpha}_{\langle[k]\rangle})\right].$$

Considering the expectation of this quantity, we are now ready to use Assumption 1 on the *local geometric improvement* of the inner procedure. We have

$$\mathbf{E}[D(\boldsymbol{\alpha}^{(t+1)}) - D(\boldsymbol{\alpha}^{(t)}) \,|\, \boldsymbol{\alpha}^{(t)}] \geq \frac{1}{K}\sum_{k=1}^K \mathbf{E}[\varepsilon_{D,k}(\boldsymbol{\alpha}^{(t)}) - \varepsilon_{D,k}(\boldsymbol{\alpha}^{(t)} + \Delta\boldsymbol{\alpha}_{\langle[k]\rangle}) \,|\, \boldsymbol{\alpha}^{(t)}]$$
$$\overset{(4)}{\geq} \frac{1}{K}(1 - \Theta)\sum_{k=1}^K \varepsilon_{D,k}(\boldsymbol{\alpha}^{(t)}).$$

It remains to bound $\sum_{k=1}^K \varepsilon_{D,k}(\boldsymbol{\alpha}^{(t)})$.

$$\sum_{k=1}^K \varepsilon_{D,k}(\boldsymbol{\alpha}^{(t)}) \overset{(3)}{=} \max_{\hat{\boldsymbol{\alpha}} \in \mathbb{R}^n} \left\{\sum_{k=1}^K \left[D((\boldsymbol{\alpha}_{[1]}, \ldots, \hat{\boldsymbol{\alpha}}_{[k]}, \ldots, \boldsymbol{\alpha}_{[K]})) - D((\boldsymbol{\alpha}_{[1]}, \ldots, \boldsymbol{\alpha}_{[k]}, \ldots, \boldsymbol{\alpha}_{[K]}))\right]\right\}$$
$$\overset{(2)}{=} \max_{\hat{\boldsymbol{\alpha}} \in \mathbb{R}^n} \left\{\frac{1}{n}\sum_{i=1}^n (-\ell_i^*(-\hat{\alpha}_i) + \ell_i^*(-\alpha_i^{(t)}))\right.$$
$$\left. + \frac{\lambda}{2}\sum_{k=1}^K \left[-\|A\boldsymbol{\alpha}^{(t)} + A_{[k]}(\hat{\boldsymbol{\alpha}}_{[k]} - \boldsymbol{\alpha}_{[k]}^{(t)})\|^2 + \|A\boldsymbol{\alpha}^{(t)}\|^2\right]\right\}$$
$$= \max_{\hat{\boldsymbol{\alpha}} \in \mathbb{R}^n} \left\{D(\hat{\boldsymbol{\alpha}}) - D(\boldsymbol{\alpha}^{(t)}) + \frac{\lambda}{2}\|A\hat{\boldsymbol{\alpha}}\|^2 - \frac{\lambda}{2}\|A\boldsymbol{\alpha}^{(t)}\|^2\right.$$
$$\left. + \frac{\lambda}{2}\sum_{k=1}^K \left[-\|A\boldsymbol{\alpha}^{(t)} + A_{[k]}(\hat{\boldsymbol{\alpha}}_{[k]} - \boldsymbol{\alpha}_{[k]}^{(t)})\|^2 + \|A\boldsymbol{\alpha}^{(t)}\|^2\right]\right\}$$
$$= \max_{\hat{\boldsymbol{\alpha}} \in \mathbb{R}^n} \left\{D(\hat{\boldsymbol{\alpha}}) - D(\boldsymbol{\alpha}^{(t)}) + \frac{\lambda}{2}\left[\|A\hat{\boldsymbol{\alpha}}\|^2 - \|A\boldsymbol{\alpha}^{(t)}\|^2\right]\right.$$
$$\left. + \frac{\lambda}{2}\left[-2(A\boldsymbol{\alpha}^{(t)})^T A(\hat{\boldsymbol{\alpha}} - \boldsymbol{\alpha}^{(t)}) - \sum_{k=1}^K \|A_{[k]}(\hat{\boldsymbol{\alpha}}_{[k]} - \boldsymbol{\alpha}_{[k]}^{(t)})\|^2\right]\right\}$$

$$= \max_{\hat{\boldsymbol{\alpha}} \in \mathbb{R}^n} \left\{ D(\hat{\boldsymbol{\alpha}}) - D(\boldsymbol{\alpha}^{(t)}) + \tfrac{\lambda}{2} \left[ \|A\hat{\boldsymbol{\alpha}}\|^2 + \|A\boldsymbol{\alpha}^{(t)}\|^2 - 2(A\boldsymbol{\alpha}^{(t)})^T A\hat{\boldsymbol{\alpha}} \right] \right.$$

$$\left. + \tfrac{\lambda}{2} \left[ -\sum_{k=1}^K \|A_{[k]}(\hat{\boldsymbol{\alpha}}_{[k]} - \boldsymbol{\alpha}_{[k]}^{(t)})\|^2 \right] \right\}$$

$$= \max_{\hat{\boldsymbol{\alpha}} \in \mathbb{R}^n} \left\{ D(\hat{\boldsymbol{\alpha}}) - D(\boldsymbol{\alpha}^{(t)}) + \tfrac{\lambda}{2} \left[ \|A(\boldsymbol{\alpha}^{(t)} - \hat{\boldsymbol{\alpha}})\|^2 \right] \right.$$

$$\left. + \tfrac{\lambda}{2} \left[ -\sum_{k=1}^K \|A_{[k]}(\hat{\boldsymbol{\alpha}}_{[k]} - \boldsymbol{\alpha}_{[k]}^{(t)})\|^2 \right] \right\}$$

$$\geq \max_{\hat{\boldsymbol{\alpha}} \in \mathbb{R}^n} \left\{ D(\hat{\boldsymbol{\alpha}}) - D(\boldsymbol{\alpha}^{(t)}) - \tfrac{\sigma}{2\lambda n^2} \|\hat{\boldsymbol{\alpha}} - \boldsymbol{\alpha}^{(t)}\|^2 \right\},$$

by the definition (7) of the complexity parameter $\sigma$. Now, we can conclude the bound as follows:

$$\sum_{k=1}^K \varepsilon_{D,k}(\boldsymbol{\alpha}^{(t)}) \geq \max_{\hat{\boldsymbol{\alpha}} \in \mathbb{R}^n} \left\{ D(\hat{\boldsymbol{\alpha}}) - D(\boldsymbol{\alpha}^{(t)}) - \tfrac{\sigma}{2\lambda n^2} \|\hat{\boldsymbol{\alpha}} - \boldsymbol{\alpha}^{(t)}\|^2 \right\}$$

$$\geq \max_{\eta \in [0,1]} \left\{ D(\eta \boldsymbol{\alpha}^* + (1-\eta)\boldsymbol{\alpha}^{(t)}) - D(\boldsymbol{\alpha}^{(t)}) - \tfrac{1}{2} \tfrac{\sigma \eta^2}{\lambda n^2} \|\boldsymbol{\alpha}^* - \boldsymbol{\alpha}^{(t)}\|^2 \right\}$$

$$\geq \max_{\eta \in [0,1]} \left\{ \eta D(\boldsymbol{\alpha}^*) + (1-\eta) D(\boldsymbol{\alpha}^{(t)}) - D(\boldsymbol{\alpha}^{(t)}) \right.$$

$$\left. + \tfrac{\gamma \eta (1-\eta)}{2n} \|\boldsymbol{\alpha}^* - \boldsymbol{\alpha}^{(t)}\|^2 - \tfrac{1}{2} \tfrac{\sigma \eta^2}{\lambda n^2} \|\boldsymbol{\alpha}^* - \boldsymbol{\alpha}^{(t)}\|^2 \right\}$$

$$\geq \max_{\eta \in [0,1]} \left\{ \eta(D(\boldsymbol{\alpha}^*) - D(\boldsymbol{\alpha}^{(t)})) + \tfrac{\eta}{2n} \left( \gamma(1-\eta) - \tfrac{\sigma \eta}{\lambda n} \right) \|\boldsymbol{\alpha}^* - \boldsymbol{\alpha}^{(t)}\|^2 \right\}.$$

Choosing $\eta^* := \frac{\lambda n \gamma}{\sigma + \lambda n \gamma} \in [0,1]$ gives

$$\sum_{k=1}^K \varepsilon_{D,k}(\boldsymbol{\alpha}^{(t)}) \geq \tfrac{\lambda n \gamma}{\sigma + \lambda n \gamma} (D(\boldsymbol{\alpha}^*) - D(\boldsymbol{\alpha}^{(t)})).$$

Therefore, we have

$$\mathbf{E}[D(\boldsymbol{\alpha}^{(t+1)}) - D(\boldsymbol{\alpha}^*) + D(\boldsymbol{\alpha}^*) - D(\boldsymbol{\alpha}^{(t)}) \mid \boldsymbol{\alpha}^{(t)}] \geq (1-\Theta)\tfrac{1}{K}\tfrac{\lambda n \gamma}{\sigma + \lambda n \gamma}(D(\boldsymbol{\alpha}^*) - D(\boldsymbol{\alpha}^{(t)})),$$

$$\mathbf{E}[D(\boldsymbol{\alpha}^*) - D(\boldsymbol{\alpha}^{(t+1)}) \mid \boldsymbol{\alpha}^{(t)}] \leq \left[ 1 - (1-\Theta)\tfrac{1}{K}\tfrac{\lambda n \gamma}{\sigma + \lambda n \gamma} \right](D(\boldsymbol{\alpha}^*) - D(\boldsymbol{\alpha}^{(t)})).$$

This implies the claim (6) of the theorem. $\qquad\square$

## B  Proof of Proposition 1 – Local Improvement of the Inner Optimizer

### B.1  Decomposition of the Duality Structure over the Blocks of Coordinates

The core concept in our new analysis technique is the following primal-dual structure on each local coordinate block. Using this, we will show below that all steps of LOCALDUALMETHOD can be interpreted as performing coordinate ascent steps on the *global* dual objective function $D(\boldsymbol{\alpha})$, with the coordinates changes restricted to the $k$-th block, as follows.

For concise notation, we write $\{\mathcal{I}_k\}_{k=1}^K$ for the partition of data indices $\{1, 2, \ldots, n\}$ between the workers, such that $|\mathcal{I}_k| = n_k$. In other words, each set $\mathcal{I}_k$ consists of the indices of the $k$-th block of coordinates (those with the dual variables $\boldsymbol{\alpha}_{[k]}$ and datapoints $A_{[k]}$, as available on the $k$-th worker).

Let us define the *local dual problem* as

$$\max_{\boldsymbol{\alpha}_{[k]} \in \mathbb{R}^{n_k}} \left[ D_k(\boldsymbol{\alpha}_{[k]}; \overline{\boldsymbol{w}}) := -\tfrac{\lambda}{2} \|\overline{\boldsymbol{w}} + A_{[k]}\boldsymbol{\alpha}_{[k]}\|^2 - \tfrac{1}{n} \sum_{i \in \mathcal{I}_k} \ell_i^*(-\alpha_i) + \tfrac{\lambda}{2}\|\overline{\boldsymbol{w}}\|^2 \right]. \qquad (8)$$

The definition is valid for any fixed vector $\overline{\boldsymbol{w}}$. Here the reader should think of $\overline{\boldsymbol{w}}$ as representing the status of those dual variables $\alpha_i$ which are *not* part of the active block $k$. The idea is the following: For the choice of $\overline{\boldsymbol{w}} := \sum_{k' \neq k} A_{[k']}\boldsymbol{\alpha}_{[k']}^{(0)}$, it turns out that the local dual objective, as a function of the local variables $\boldsymbol{\alpha}_{[k]}$, is identical the global (dual) objective, up to a constant independent of $\boldsymbol{\alpha}_{[k]}$, or formally

$$D_k(\boldsymbol{\alpha}_{[k]}; \overline{\boldsymbol{w}}) = D(\boldsymbol{\alpha}_{[1]}^{(0)}, \ldots, \boldsymbol{\alpha}_{[k]}, \ldots, \boldsymbol{\alpha}_{[K]}^{(0)}) + C' \qquad \forall\, \boldsymbol{\alpha}_{[k]} \in \mathbb{R}^{n_k}.$$

The following proposition observes that the defined local problem has a very similar duality structure as the original problem (1) with its dual (2).

**Proposition 4.** *For $\overline{w} \in \mathbb{R}^d$, let us define the "local" primal problem on the $k$-th block as*

$$\min_{\boldsymbol{w}_k \in \mathbb{R}^d} \left[ \; P_k(\boldsymbol{w}_k; \overline{\boldsymbol{w}}) := \frac{1}{n} \sum_{i \in \mathcal{I}_k} \ell_i((\overline{\boldsymbol{w}} + \boldsymbol{w}_k)^T \boldsymbol{x}_i) + \frac{\lambda}{2} \|\boldsymbol{w}_k\|^2 \; \right]. \tag{9}$$

*Then the dual of this formulation (with respect to the variable $\boldsymbol{w}_k$) is given by the local dual problem (8) for the $k$-th coordinate block.*

*Proof.* The dual of this problem is derived by plugging in the definition of the conjugate function $\ell_i((\overline{\boldsymbol{w}} + \boldsymbol{w}_k)^T \boldsymbol{x}_i) = \max_{\alpha_i} -\alpha_i (\overline{\boldsymbol{w}} + \boldsymbol{w}_k)^T \boldsymbol{x}_i - \ell_i^*(-\alpha_i)$, which gives

$$\min_{\boldsymbol{w}_k \in \mathbb{R}^d} \quad \frac{1}{n} \sum_{i \in \mathcal{I}_k} \max_{\alpha_i} \left( - \alpha_i (\overline{\boldsymbol{w}} + \boldsymbol{w}_k)^T \boldsymbol{x}_i - \ell_i^*(-\alpha_i) \right) + \frac{\lambda}{2} \|\boldsymbol{w}_k\|^2$$

$$= \frac{1}{n} \sum_{i \in \mathcal{I}_k} \max_{\alpha_i} -\ell_i^*(-\alpha_i) + \min_{\boldsymbol{w}_k \in \mathbb{R}^d} \left[ -\alpha_i (\overline{\boldsymbol{w}} + \boldsymbol{w}_k)^T \boldsymbol{x}_i + \frac{\lambda}{2} \|\boldsymbol{w}_k\|^2 \right]$$

$$= \max_{\boldsymbol{\alpha} \in \mathbb{R}^{n_k}} \frac{1}{n} \sum_{i \in \mathcal{I}_k} -\ell_i^*(-\alpha_i) + \min_{\boldsymbol{w}_k \in \mathbb{R}^d} \left[ -\frac{1}{n} \sum_{i \in \mathcal{I}_k} \alpha_i (\overline{\boldsymbol{w}} + \boldsymbol{w}_k)^T \boldsymbol{x}_i + \frac{\lambda}{2} \|\boldsymbol{w}_k\|^2 \right].$$

The first-order optimality condition for $\boldsymbol{w}_k$, by setting its derivative to zero in the inner minimization, can be written as

$$\boldsymbol{w}_k^* = \frac{1}{\lambda n} \sum_{i \in \mathcal{I}_k} \alpha_i \boldsymbol{x}_i = A_{[k]} \boldsymbol{\alpha}_{[k]}$$

plugging this back in, we have that the inner minimization becomes

$$- \lambda \frac{1}{\lambda n} \sum_{i \in \mathcal{I}_k} \alpha_i (\overline{\boldsymbol{w}} + \boldsymbol{w}_k^*)^T \boldsymbol{x}_i + \frac{\lambda}{2} \|\boldsymbol{w}_k^*\|^2$$

$$= - \lambda (\overline{\boldsymbol{w}} + \boldsymbol{w}_k^*)^T \boldsymbol{w}_k^* + \frac{\lambda}{2} \|\boldsymbol{w}_k^*\|^2$$

Writing the resulting full problem, we obtain precisely the local dual problem (8) for the $k$-th coordinate block. $\qquad\square$

Using these local subproblems, we have the following nice structure of local and global duality gaps:

$$g_k(\boldsymbol{\alpha}) := P_k(\boldsymbol{w}_k; \overline{\boldsymbol{w}}) - D_k(\boldsymbol{\alpha}_{[k]}; \overline{\boldsymbol{w}}) \qquad \text{and} \qquad g(\boldsymbol{\alpha}) := P(\boldsymbol{w}(\boldsymbol{\alpha})) - D(\boldsymbol{\alpha})$$

Here the contributions of all blocks except the active one are collected in $\overline{\boldsymbol{w}} := \boldsymbol{w} - \boldsymbol{w}_k$. Recall that we defined $\boldsymbol{w}_k := A_{[k]} \boldsymbol{\alpha}_{[k]}$ and $\boldsymbol{w} := A\boldsymbol{\alpha}$, so that $\boldsymbol{w} = \sum_{k=1}^{K} \boldsymbol{w}_k$.

## B.2 Local Convergence of LOCALSDCA

This section is just a small modification of results obtained in [13].

Observe that the coordinate step performed by an iteration of LOCALSDCA can equivalently be written as

$$\Delta\alpha_i^* = \Delta\alpha_i^* \big( \boldsymbol{\alpha}_{[k]}^{(h-1)}, \overline{\boldsymbol{w}} \big) := \underset{\Delta\alpha_i}{\operatorname{argmax}} \left( D_k(\boldsymbol{\alpha}_{[k]}^{(h-1)} + e_i \Delta\alpha_i; \overline{\boldsymbol{w}}) \right). \tag{10}$$

For a vector $\overline{\boldsymbol{w}} \in \mathbb{R}^d$. In other words, the step will optimize one of the local coordinates with respect to the local dual objective, which is identical to the global dual objective when the coordinates of the other blocks are kept fixed, as we have seen in the previous subsection.

**Lemma 5.** *Assume that $\ell_i^*$ is $\gamma$-strongly convex (where $\gamma \geq 0$). Then for all iterations $h$ of* LO-CALSDCA *and any $s \in [0,1]$ we have*

$$\mathbf{E}[D_k(\boldsymbol{\alpha}_{[k]}^{(h)}; \overline{\boldsymbol{w}}) - D_k(\boldsymbol{\alpha}_{[k]}^{(h-1)}; \overline{\boldsymbol{w}})] \geq \frac{s}{n_k}\left(P_k(\boldsymbol{w}_k^{(h-1)}; \overline{\boldsymbol{w}}) - D_k(\boldsymbol{\alpha}_{[k]}^{(h-1)}; \overline{\boldsymbol{w}})\right) - \frac{s^2}{2\lambda n^2} G^{(h)}, \quad (11)$$

*where $G^{(h)} := \frac{1}{n_k}\sum_{i \in \mathcal{I}_k}\left(\|\boldsymbol{x}_i\|^2 - \frac{\lambda n \gamma(1-s)}{s}\right)(u_i^{(h)} - \alpha_i^{(h-1)})^2$, $-u_i^{(h-1)} \in \partial\ell_i(\boldsymbol{x}_i^T \boldsymbol{w}^{(h-1)})$ and $\boldsymbol{w}^{(h-1)} = \overline{\boldsymbol{w}} + A_{[k]}\boldsymbol{\alpha}_{[k]}^{(h-1)}$.*

*Proof.*

$$n\left[D_k(\boldsymbol{\alpha}_{[k]}^{(h)}; \overline{\boldsymbol{w}}) - D_k(\boldsymbol{\alpha}_{[k]}^{(h-1)}; \overline{\boldsymbol{w}})\right]$$

$$= \underbrace{-\ell_i^*(-\alpha_i^{(h)}) - \frac{\lambda n}{2}\|\boldsymbol{w}^{(h)}\|^2}_{A} - \underbrace{\left(-\ell_i^*(-\alpha_i^{(h-1)}) - \frac{\lambda n}{2}\|\boldsymbol{w}^{(h-1)}\|^2\right)}_{B}.$$

By the definition of the update (10) we have for all $s \in [0,1]$ that

$$A = \max_{\Delta\alpha_i} -\ell_i^*(-\alpha_i^{(h-1)} - \Delta\alpha_i) - \frac{\lambda n}{2}\|\boldsymbol{w}^{(h-1)} + \frac{1}{\lambda n}\Delta\alpha_i \boldsymbol{x}_i\|^2$$

$$\geq -\ell_i^*(-\alpha_i^{(h-1)} - s(u_i^{(h)} - \alpha_i^{(h-1)})) - \frac{\lambda n}{2}\|\boldsymbol{w}^{(h-1)} + \frac{1}{\lambda n}s(u_i^{(h)} - \alpha_i^{(h-1)})\boldsymbol{x}_i\|^2.$$

From strong convexity we have

$$\ell_i^*\left(-\alpha_i^{(h-1)} - s(u_i^{(h)} - \alpha_i^{(h-1)})\right)$$

$$\leq s\ell_i^*(-u_i^{(h)}) + (1-s)\ell_i^*(-\alpha_i^{(h-1)}) - \frac{\gamma}{2}s(1-s)(u_i^{(h)} - \alpha_i^{(h-1)})^2. \quad (12)$$

Hence

$$A \overset{(12)}{\geq} -s\ell_i^*(-u_i^{(h)}) - (1-s)\ell_i^*(-\alpha_i^{(h-1)}) + \frac{\gamma}{2}s(1-s)(u_i^{(h)} - \alpha_i^{(h-1)})^2$$

$$- \frac{\lambda n}{2}\|\boldsymbol{w}^{(h-1)} + \frac{1}{\lambda n}s(u_i^{(h)} - \alpha_i^{(h-1)})\boldsymbol{x}_i\|^2$$

$$= \underbrace{-s(\ell_i^*(-u_i^{(h)}) + u_i^{(h)}\boldsymbol{x}_i^T \boldsymbol{w}^{(h-1)})}_{s\ell(\boldsymbol{x}_i^T \boldsymbol{w}^{(h-1)})} + \underbrace{(-\ell_i^*(-\alpha_i^{(h-1)}) - \frac{\lambda n}{2}\|\boldsymbol{w}^{(h-1)}\|^2)}_{B}$$

$$+ \frac{s}{2}\left(\gamma(1-s) - \frac{1}{\lambda n}s\|\boldsymbol{x}_i\|^2\right)(u_i^{(h)} - \alpha_i^{(h-1)})^2 + s(\ell_i^*(-\alpha_i^{(h-1)}) + \alpha_i^{(h-1)}\boldsymbol{x}_i^T \boldsymbol{w}^{(h-1)}.$$

Therefore

$$A - B \geq s\left[\ell(\boldsymbol{x}_i^T \boldsymbol{w}^{(h-1)}) + \ell_i^*(-\alpha_i^{(h-1)}) + \alpha_i^{(h-1)}\boldsymbol{x}_i^T \boldsymbol{w}^{(h-1)}\right.$$

$$\left. + \frac{1}{2}\left(\gamma(1-s) - \frac{1}{\lambda n}s\|\boldsymbol{x}_i\|^2\right)(u_i^{(h)} - \alpha_i^{(h-1)})^2\right]. \quad (13)$$

Recall that our above definition of the local pair of primal and dual problems gives

$$P_k(\boldsymbol{w}_k; \overline{\boldsymbol{w}}) - D_k(\boldsymbol{\alpha}_{[k]}; \overline{\boldsymbol{w}}) = \frac{1}{n}\sum_{i \in \mathcal{I}_k}\left(\ell_i((\overline{\boldsymbol{w}} + \boldsymbol{w}_k)^T \boldsymbol{x}_i) + \ell_i^*(-\alpha_i) + \alpha_i(\overline{\boldsymbol{w}} + \boldsymbol{w}_k)^T \boldsymbol{x}_i\right).$$

where $\boldsymbol{w}_k := A_{[k]}\boldsymbol{\alpha}_{[k]}$.

If we take the expectation of (13) we obtain

$$\frac{1}{s}\,\mathbf{E}[A-B] \geq \frac{n}{n_k}\underbrace{\frac{1}{n}\sum_{i\in\mathcal{I}_k}\left[\ell(\boldsymbol{x}_i^T\boldsymbol{w}^{(h-1)}) + \ell_i^*(-\alpha_i^{(h-1)}) + \alpha_i^{(h-1)}\boldsymbol{x}_i^T\boldsymbol{w}^{(h-1)}\right]}_{P_k(w_k^{(h-1)};\overline{\boldsymbol{w}}) - D_k(\boldsymbol{\alpha}_{[k]}^{(h-1)};\overline{\boldsymbol{w}})}$$

$$-\frac{s}{2\lambda n}\underbrace{\frac{1}{n_k}\sum_{i\in\mathcal{I}_k}\left(\|\boldsymbol{x}_i\|^2 - \frac{\lambda n\gamma(1-s)}{s}\right)(u_i^{(h)} - \alpha_i^{(h-1)})^2}_{G^{(h)}}.$$

Therefore, we have obtained the claimed improvement bound

$$\frac{n}{s}\,\mathbf{E}[D_k(\boldsymbol{\alpha}_{[k]}^{(h)};\overline{\boldsymbol{w}}) - D_k(\boldsymbol{\alpha}_{[k]}^{(h-1)};\overline{\boldsymbol{w}})] \geq \frac{n}{n_k}(P_k(\boldsymbol{w}_k^{(h-1)};\overline{\boldsymbol{w}}) - D_k(\boldsymbol{\alpha}_{[k]}^{(h-1)};\overline{\boldsymbol{w}})) - \frac{s}{2\lambda n}G^{(h)}.$$

$\square$

**Proposition' 1.** *Assume the loss functions $\ell_i$ are $(1/\gamma)$-smooth. Then for using* LOCALSDCA, *Assumption 1 holds with*

$$\Theta = \left(1 - \frac{\lambda n\gamma}{1+\lambda n\gamma}\frac{1}{\tilde{n}}\right)^H. \tag{14}$$

*where $\tilde{n} := \max_k n_k$ is the size of the largest block of coordinates.*

**Lemma 6** (Local Convergence on the Subproblem). *For any $\boldsymbol{\alpha}_{[k]}^{(0)} \in \mathbb{R}^{n_k}$ and $\overline{\boldsymbol{w}} \in \mathbb{R}^d$ let us define*

$$\boldsymbol{\alpha}_{[k]}^{(*)} := \operatorname*{argmax}_{\boldsymbol{\alpha}_{[k]}\in\mathbb{R}^{n_k}} D_k(\boldsymbol{\alpha}_{[k]};\overline{\boldsymbol{w}}). \tag{15}$$

*If* LOCALSDCA *is used for $H$ iterations on block $k$, then*

$$\mathbf{E}\left[D_k(\boldsymbol{\alpha}_{[k]}^{(*)};\overline{\boldsymbol{w}}) - D_k(\boldsymbol{\alpha}_{[k]}^{(H)};\overline{\boldsymbol{w}})\right] \leq \left(1 - \frac{s}{n_k}\right)^H\left(D_k(\boldsymbol{\alpha}_{[k]}^{(*)};\overline{\boldsymbol{w}}) - D_k(\boldsymbol{\alpha}_{[k]}^{(0)};\overline{\boldsymbol{w}})\right). \tag{16}$$

*Proof.* We will use Lemma 5 with $s := \frac{\lambda n\gamma}{1+\lambda n\gamma} \in [0,1]$. Because $\|\boldsymbol{x}_i\| \leq 1$, we have $G^{(h)} \leq 0$. Therefore

$$\mathbf{E}\left[D_k(\boldsymbol{\alpha}_{[k]}^{(h)};\overline{\boldsymbol{w}}) - D_k(\boldsymbol{\alpha}_{[k]}^{(h-1)};\overline{\boldsymbol{w}})\right] \geq \frac{s}{n_k}\left(P_k(\boldsymbol{w}_k^{(h-1)};\overline{\boldsymbol{w}}) - D_k(\boldsymbol{\alpha}_{[k]}^{(h-1)};\overline{\boldsymbol{w}})\right). \tag{17}$$

Following the proof of Theorem 5 in [13] we obtain the claimed bound. $\square$

Now, to prove Proposition 1 it is enough to observe that for fixed $k$ and any $\boldsymbol{\alpha} = (\boldsymbol{\alpha}_{[1]},\ldots,\boldsymbol{\alpha}_{[K]}) \in \mathbb{R}^n$, assuming we define $\overline{\boldsymbol{w}} = \sum_{k'\neq k} A_{[k']}\boldsymbol{\alpha}_{[k']}$, it holds that

$$\varepsilon_{D,k}(\boldsymbol{\alpha}) \overset{(3)}{=} \max_{\hat{\boldsymbol{\alpha}}_{[k]}\in\mathbb{R}^{n_k}} D((\boldsymbol{\alpha}_{[1]},\ldots,\hat{\boldsymbol{\alpha}}_{[k]},\ldots,\boldsymbol{\alpha}_{[K]})) - D((\boldsymbol{\alpha}_{[1]},\ldots,\boldsymbol{\alpha}_{[k]},\ldots,\boldsymbol{\alpha}_{[K]}))$$

$$= D((\boldsymbol{\alpha}_{[1]},\ldots,\boldsymbol{\alpha}_{[k]}^{(*)},\ldots,\boldsymbol{\alpha}_{[K]})) - D((\boldsymbol{\alpha}_{[1]},\ldots,\boldsymbol{\alpha}_{[k]},\ldots,\boldsymbol{\alpha}_{[K]}))$$

$$= D_k(\boldsymbol{\alpha}_{[k]}^{(*)};\overline{\boldsymbol{w}}) - D_k(\boldsymbol{\alpha}_{[k]};\overline{\boldsymbol{w}}),$$

where $\boldsymbol{\alpha}_{[k]}^{(*)}$ is defined by (15).

## C  Proof of Lemma 3 – The Problem Complexity Parameter $\sigma_{\min}$

**Lemma' 3.** *If $K = 1$ then $\sigma_{\min} = 0$. For any $K \geq 1$, when assuming $\|\boldsymbol{x}_i\| \leq 1 \ \forall i$, we have*

$$0 \leq \sigma_{\min} \leq \tilde{n}.$$

*Moreover, if datapoints between different workers are orthogonal, i.e. $(A^T A)_{i,j} = 0 \ \forall i,j$ such that $i$ and $j$ do not belong to the same part, then $\sigma_{\min} = 0$.*

*Proof.* If $K = 1$ then $\boldsymbol{\alpha}_{[K]} \equiv \boldsymbol{\alpha}$ and hence $\sigma_{\min} = 0$. For a non-trivial case, when $K > 1$ we have

$$
\sigma_{\min} \overset{(7)}{=} \max_{\boldsymbol{\alpha} \in \mathbb{R}^n} \lambda^2 n^2 \frac{\sum_{k=1}^K \|A_{[k]}\boldsymbol{\alpha}_{[k]}\|^2 - \|A\boldsymbol{\alpha}\|^2}{\|\boldsymbol{\alpha}\|^2}
$$

$$
= \max_{\boldsymbol{\alpha} \in \mathbb{R}^n} \frac{\sum_{k=1}^K \|X_{[k]}\boldsymbol{\alpha}_{[k]}\|^2 - \|X\boldsymbol{\alpha}\|^2}{\|\boldsymbol{\alpha}\|^2}
$$

$$
\leq \max_{\boldsymbol{\alpha} \in \mathbb{R}^n} \frac{\sum_{k=1}^K \|X_{[k]}\boldsymbol{\alpha}_{[k]}\|^2}{\sum_{k=1}^K \|\boldsymbol{\alpha}_{[k]}\|^2} \leq \max_{\boldsymbol{\alpha} \in \mathbb{R}^n} \frac{\sum_{k=1}^K \tilde{n}\|\boldsymbol{\alpha}_{[k]}\|^2}{\sum_{k=1}^K \|\boldsymbol{\alpha}_{[k]}\|^2} \leq \tilde{n},
$$

where we have used the definition of the rescaled data matrix $A = \frac{1}{\lambda n} X$. The last inequality follows from the fact that since the datapoints have bounded norm $\|\boldsymbol{x}_i\| \leq 1$, so that for all parts $k \in \{1, \ldots, K\}$ we have
$\max_{\boldsymbol{\alpha}_{[k]} \in \mathbb{R}^{n_k}} \frac{\|X_{[k]}\boldsymbol{\alpha}_{[k]}\|^2}{\|\boldsymbol{\alpha}_{[k]}\|^2} = \|X_{[k]}\|_{\text{op}}^2 \leq \|X_{[k]}\|_{\text{frob}}^2 \leq n_k \leq \tilde{n}$, where $\tilde{n} := \max_k n_k$.

The case when $A^T A$ is block-diagonal (with respect to the partition) is trivial. $\qquad \square$