[Reviews · NeurIPS 2014]

Submitted by Assigned_Reviewer_20

The paper describes a distributed optimization scheme following the prevalent data-partitioning approach. Different from recent works, the authors suggest to use a primal-dual solver as the optimizer on each data partition. Their analysis shows that doing so allows the distributed method to retain the convergence of the inner solver and that the number of machines used serves as a pure speedup, modulo communication cost.

The paper is well written and easy to follow and the argumentation is convincing.

However, what I missed most when reading it was an honest comparison with primal batch methods that can equally be implemented in a distributed system. I alluded to some of the similarities in the slanted summary I put above. But an actual theoretical treatment of those methods would do this paper good. Especially considering that SGD is now wildly understood as a poor fit for environments such as MapReduce.

Nitpicks below:

Page 1, line 41: "any" seems to be the wrong word here
Page 2, line 71: It should be made more clear what you mean by "iteration" here, as the MapReduce paragdigm only presents one record at a time to the map function.
Page 2, Line 107: Here, you made it somewhat easy to find out the identity of the authors.
Summary: An OK paper that provides an analysis of the data-parallel distribution paradigm when using primal-dual optimizers on each data partition. What's missing is a comparison with primal methods of similar per-iteration communications cost.

Submitted by Assigned_Reviewer_24

summary:
The paper introduces a new approach to parallel coordinate descent for primal-dual linear methods. The dual problem is split into blocks of variables and each of these blocks is solved independently with an arbitrary optimizer. At the end of each iteration the (weighted) mean of the partial solutions is computed and shared among the nodes. The rate of convergence is analysed and it is shown that the algorithm converges with a linear rate when the used optimizer of the block does. Good results are shown on a set of problems.

Quality:
The paper is overall of high quality. The proofs appear to be correct and the experiments are reasonably well executed. However, I would have liked to see a few more results: On the one hand there is an effect of how the data points are distributed among the nodes and on the other hand there is a strong dependency on the strength of regularization. With small regularization, a big H will lead to a higher conflict between solutions, such that the convergence rate might be very small and a lot of computation is wasted.

clarity:
The paper is overall well written and structured. However, there are a few notational problems and typos:

Section 4, Setup: It appears to me that alpha*_k is used ambiguously. first in epsilon_{D,k} as the optimal solution when only the k-th block is changed and then in epsilon_D as a notion of the global optimum. This is very confusing.

Definition 1: It would be easier to state this in terms of the linear convergence rate, as "local geometric power" seems to not be very well known.
Lemma 2: "(1/gamma)-smooth" is at least not known to me and it is also not defined in the appendix. I would have liked to see a definition for this.

Proposition 4: beta_K is given two different values.

Originality:
This paper merges several line of thoughts into one easy algorithm. What it distinguishes it from other approaches is the convergence proof. (I am, however, not an expert in this type of algorithms and thus I might miss an important reference.)
Summary: The paper is overall a novel contribution and well written. However there are a few minor notational issues and the experimental section might be expanded to have a better understanding of the behaviour of the algorithm.

Submitted by Assigned_Reviewer_38

The paper describes a parallel variant of SDCA algorithm which each machine executes SDCA updates using local parameters and these local parameters are periodically averaged. Theorem 3 provides the convergence guarantee of the algorithm.

Quality: Unfortunately, Theorem 3 is not very interesting since the bound is a function of T/K; as T should be a linear function of K at best, there will be no speed-up by increasing K, the number of workers. What this means is that if we use beta_K = 1, which is the only case the theorem covers, then the algorithm will become more and more conservative as we increase the number of machines, and compared to the case of single-thread execution the algorithm will not be any faster. In practice we could make the algorithm more aggressive by using beta_K > 1, but the paper does not provide theoretical support. The empirical comparison against other methods seems motivating, but unfortunately an experiment on scaling behavior of the algorithm as a function of K is missing, which I believe is critically important considering Theorem 3 does not guarantee speed-up.

Clarity: Lemmas and theorems in the paper are well-organized and easy to follow, since authors have done a good job in providing intuitive explanation of the proof.

Originality: The proposed algorithm is a generalization of a popular heuristic which local parameters are periodically averaged, and the proof does seem to provide new techniques or insights; the proof of Lemma 2 does not depart much from the original proof for SDCA, and then the main theorem (Theorem 3) is just an application of Jensen's on top of Lemma 2. This is why Theorem 3 is weak and does not provide any speed-up guarantee; other parallel algorithms such as 11 or 21 departs from the proof of single-thread case by taking account of correlation between data points and therefore are able to provide speed-up guarantees. I believe it should be necessary as well for parallel SDCA algorithm to consider correlation between data points, if one wants to achieve a nontrivial convergence guarantee.

Significance: The paper provides a nice insight that H, the period of synchronization, might have to be a function of C, the local convergence speed. However, this insight is based on a loose convergence guarantee (Theorem 3) which does not provide any parallelization speedup. Even worse, since C is not known a priori, one cannot choose H without actually executing the algorithm.
Summary: The analysis of the proposed algorithm is not very useful because it does not guarantee any parallelization speed-up in terms of the number of workers.
Author Feedback
Author rebuttal: We thank the reviewers for their thoughtful comments. We first address concerns shared by all 3 reviewers, and then respond to specific comments below.

1) Significance of Convergence Rate:

It is indeed very important to compare our new framework and theory to what is already known for mini-batch versions of SGD and SDCA. It is true that, in terms of outer iterations T, our new rate is not better than the known rates for distributed versions of these algorithms. However, our framework uses vastly less communication: Every round sends K data examples, while analogous mini-batch methods (and also [11] and [21]) send H*K examples, where H is in the order of the (local) data size. Our main contribution is exactly to reduce communication by this large factor H, while maintaining the convergence rate. Unfortunately, we feel that the reviewers have missed this main difference in their assessment of the paper, particularly in addressing the impact and originality of the work.

The quantity H - the number of data examples touched before a single vector is communicated - can be freely chosen to balance the trade-off between communication and local computation, and is often set to be on the order n in realistic settings. We are not aware of any proven rates for a scheme with equally restricted communication. (See also the comment on single round communication schemes below in response to Reviewer_38.)

2) Originality & Correlation:

Correlation in the data between the machines is indeed interesting. In fact, this has been the main obstacle preventing previous work from showing convergence rates (the only existing analysis [2] was for data orthogonal between workers). It is natural to wonder if further speedups are possible by improving data partitioning. However, this hope was just very recently dismissed, as at most a speedup of 2 can be expected (even for communication inefficient mini-batch coordinate descent), as shown in [ Fercoq, O., Qu, Z., & Richtárik, P. (2014) http://arxiv.org/abs/1405.5300 ]. Therefore, the influence of data partitioning is negligible compared to our shown communication savings of H (order n). The existing rates of [11] (Shotgun), [21] (Hogwild) and [14] (Hydra) all fall into the mini-batch setting and therefore are H times slower in terms of communication.

--------------
Answers to specific reviewer questions:

@Reviewer_20

[ comparison with primal batch methods ]

Current batch primal methods, such as L-BFGS, come with no convergence guarantees even in the serial case, and in practice are not competitive with state-of-the-art online methods. In a distributed setting, if using batch methods locally instead of the online methods suggested, it is still unknown how to communicate and merge solutions in a meaningful way. This has not been addressed in the literature, except in the recent work of [Shamir, Srebro, and Zhang ICML 2014]. This work explores this direction, but requires an exact solution to a problem that is the size of the local data of each worker at each iteration. This is much more expensive than the online iterations we perform internally.

[“... SGD is now wildly understood as a poor fit for environments such as MapReduce.”]

The main point of our paper is exactly to solve this problem, by presenting the first scheme that successfully distributes online methods without the communication cost also exploding linearly in n. (See also comment (1) above.)

--------------

@Reviewer_24

[ dependence on the regularization parameter lambda ]

The advantage of coordinate descent methods over SGD is very well studied in the single machine case, see [17], and known to be robust for all levels of lambda. Our experiments show that this consistent improvement translates to the distributed setting, even in the difficult regime when lambda is as small as 1/n.

[ Notational remarks in Section 4, Definition 1, Lemma 2, and Proposition 4 ]

Many thanks for your thoughtful remarks; we will incorporate these to improve clarity.

--------------

@Reviewer_38

[“Theorem 3 is not very interesting since the bound is a function of T/K; as T should be a linear function of K at best, there will be no speed-up by increasing K…”]

We agree that the rate degrades with the workers at 1/K. Beyond what has already been discussed in comment (1) above, there are several reasons that this rate still makes significant progress as compared to state-of-the-art:

(a) As mentioned in the Section 4, the degradation with K is unavoidable, since single round communication schemes do not converge in general and their quality degrades with the number of workers. This degradation was just recently quantified [Shamir, Srebro, and Zhang ICML 2014]. The number of workers K is also typically much smaller than n, with every machine fitting > 10^6 examples.

(b) To achieve faster practical performance, schemes such as [12] scale the combination of partial results with beta_k > 1, rather than using simple averaging. As discussed in Section 4, all existing mini-batch algorithms are sensitive to the choice of this parameter beta_b, which must be found from a very large interval between 1 (for safe averaging) and b (the size of the mini-batch = K*H, for aggressive adding). These rates degrade with a factor of beta_b [12,13,14]. In contrast, in our setting, the scaling parameter is only between 1 and K, making our method much more robust. This is also reflected in Theorem 3, showing that the degradation is only with K and not with b=K*H.

(c) To emphasize, the given theory is the first for algorithms with this level of communication efficiency; our results hold without special assumptions on the data; and they apply for the much more general algorithmic framework presented here (using arbitrary internal dual optimization methods).

In practice, the best H is easily chosen as the ratio of communication/local computation for the given system.